Sniffin’ sticks smell identification test: exploring measurement properties in an adult Brazilian healthcare professionals sample

http://orcid.org/0000-0002-1191-2078 Brandão Pedro Renato de Paula 1 pedrobrandaoneurologia@gmail.com
http://orcid.org/0000-0002-1619-9179 Pereira Danilo Assis 2
http://orcid.org/0000-0001-6660-2766 Bispo Diógenes Diego de Carvalho 3
http://orcid.org/0000-0002-1004-7641 von Glehn Felipe 3
Silveira-Moriyama Laura 4
Titze-de-Almeida Simoneide Souza 5
Nakanishi Marcio 6
Saldanha-Araujo Felipe 7
Arganaraz Enrique Roberto 7 8
Ribeiro Adriana Pinheiro 9
dos Santos Agda Lima 1
Titze-de-Almeida Ricardo 5
1 Sírio-Libanês Hospital , Brasilia, DF , Brazil
2 Brazilian Institute of Neuropsychology and Cognitive Sciences , Brasilia, DF , Brazil
3 Brasilia University Hospital, University of Brasilia , Brasilia, DF , Brazil
4 State University of Campinas , Campinas , Brazil
5 Research Center for Major Themes-COVID-19 Group, University of Brasilia , Brasilia, DF , Brazil
6 School of Medicine, University of Brasilia , Brasilia, DF , Brazil
7 Faculty of Health Sciences, University of Brasilia , Brasilia, DF , Brazil
8 Lab of Molecular NeuroVirology, Faculty of Health Sciences, University of Brasilia , Brasilia, DF , Brazil
9 Armed Forces Hospital , Brasilia, DF , Brazil
Barnhart Anthony
Electronic publication date: 2025 Oct 28
Publication date: 2025
Volume: 13
Electronic Location ID: e19733
Received 2024 Nov 14; Accepted 2025 Jun 20
Copyright: © 2025 Brandão et al.
Copyright year: 2025
Copyright holder: Brandão et al.
License: This is an open access article distributed under the terms of the Creative Commons Attribution License, which permits unrestricted use, distribution, reproduction and adaptation in any medium and for any purpose provided that it is properly attributed. For attribution, the original author(s), title, publication source (PeerJ) and either DOI or URL of the article must be cited.
License URL: https://creativecommons.org/licenses/by/4.0/

Keywords: Odor identification, Sex-related differences, Sniffin’ sticks, Hyposmia, Rasch analysis

Funding: The National Council for Scientific and Technological Development (CNPq) Ministry of Education (MEC) TED/MEC n. 9249 The National Council for Scientific and Technological Development (CNPq) and the Ministry of Education (MEC) [TED/MEC n. 9249] provided an emergency response grant to support this work on strengthening laboratory infrastructure for SARS-CoV-2 epidemiological monitoring with RT-qPCR and ELISA tests. The funders had no role in study design, data collection and analysis, decision to publish, or preparation of the manuscript.

==============================
Introduction

Olfactory evaluation has gained significant attention in both neurological and otorhinolaryngological assessments, particularly following the COVID-19 pandemic where SARS-CoV-2 infection emerged as a frequent cause of chemosensory dysfunction. The Sniffin’ Sticks identification test is a widely used screening tool for olfactory function. This study aimed to investigate the psychometric properties, specifically validity evidence based on internal structure (unidimensionality) and relations to other variables (demographic effects), of the 16-item Sniffin’ Sticks odor identification test (SS-16) in a homogeneous sample of highly educated young and middle-aged Brazilian adults, while exploring item-specific demographic effects.

Methods

A prospective observational cross-sectional study was conducted on 144 highly educated adults. The SS-16 was administered using the ‘Odor-on-lines’ paradigm. Rasch analysis assessed person-item mapping and fit statistics, while multiple regression analyses determined the effects of age, sex, and education on both overall and item-specific performance.

Results

Rasch analysis supported the unidimensionality of the SS-16. Item difficulty varied, with peppermint, cinnamon, and fish being easiest to identify, while apple, turpentine, and liquorice were the most challenging. Item-level analyses revealed specific demographic influences: older age was associated with poorer identification of coffee (β = −0.05, p = 0.021) and cloves (β = −0.123, p = 0.001); female sex was associated with significantly better identification of rose (β = 1.15, p = 0.026); and higher education level positively impacted identification of cloves and anise (β = 0.122, p = 0.011). These item-specific effects suggest potential differential age-related vulnerability for certain odorants, possible hormonal influences on floral odor detection, and educational effects potentially linked to semantic processing of complex aromas. Smell identification capacity was classified as “very low” (<P3, SS-16 score ≤10), “low” (≤P9, 11 points), “average” (P9.1–P84, 12–15 points), and “high” (>P84, 16 points).

Conclusion

This study provides insights into the psychometric properties of the SS-16 in a specific sample of Brazilian adults, demonstrating both general and item-specific demographic effects on olfactory performance. The identification of item-specific influences enhances understanding of the complex interplay between biological and sociocultural factors in olfaction. While the predominance of female participants and the homogeneous high educational profile limit broader generalizability, this analysis contributes preliminary reference data for this demographic and supports the potential utility of this culturally adapted SS-16 version for clinical screening in similar Brazilian settings. Future research requires more diverse samples to establish representative Brazilian norms.

Introduction

Olfactory disorders include quantitative deficits such as hyposmia (reduced odor sensitivity) and anosmia (complete loss of smell), as well as qualitative disturbances including parosmia (distorted odor perception with a known source) and phantosmia (perception of smell in the absence of a source) (Boesveldt et al., 2017; Pellegrino et al., 2021; Patel et al., 2022). The COVID-19 pandemic highlighted these disorders when SARS-CoV-2 infection emerged as a significant cause of sudden chemosensory loss, underscoring the importance of structured smell ability testing in both otorhinolaryngology and neurology (Pellegrino et al., 2020; Las Casas Lima, Cavalcante & Leão, 2022; Doty, 2022; Parma et al., 2020; Lechien et al., 2020).

The neural mechanisms underlying olfactory processing involve complex pathways from the olfactory epithelium through the olfactory bulb to higher cortical regions Age-related decline in olfactory function, known as presbyosmia, is well-documented and stems from factors including cumulative damage to the neuroepithelium, reduced neurogenesis, and decreased central processing efficiency (Kondo et al., 2020; Pinto et al., 2015; Zhang & Wang, 2017; Rawson, 2006). Large normative studies consistently show this decline, often becoming more pronounced after the age of 60 (Hummel et al., 2007; Oleszkiewicz et al., 2019).

Sex differences in olfaction are also frequently reported, often attributed to hormonal influences, with estrogen potentially providing neuroprotective effects that may contribute to superior odor identification capacity observed in women (Doty & Cameron, 2009; Garcia-Falgueras et al., 2006; Sorokowski et al., 2019). Meta-analyses confirm that female advantage extends across detection thresholds, discrimination, and identification tasks, though effect sizes can be modest (Sorokowski et al., 2019). This advantage might be particularly evident for certain odor categories, such as floral scents like rose, potentially linked to enhanced sensitivity to specific volatile compounds such as phenylethyl alcohol (Pfabigan et al., 2022). Educational attainment can also influence olfactory performance, likely reflecting the cognitive and semantic demands inherent in odor identification tasks (Hoffmann-Hensel et al., 2017). These tasks require linking olfactory percepts to verbal labels, a process influenced by cognitive abilities, vocabulary, and semantic memory, all of which are associated with educational background (Hedner et al., 2010; Larsson et al., 2004; Larsson, Finkel & Pedersen, 2000; Kamath et al., 2024). More complex chemical stimuli or less common odors may demand more sophisticated lexical-semantic processing (Olofsson & Gottfried, 2015; Hörberg et al., 2024), making their identification potentially more susceptible to educational influences (Denervaud et al., 2021; Barea & Mansur, 2007; da Silva et al., 2004). Furthermore, age-related olfactory decline might not be uniform across all odorants; certain smells may be affected earlier or more severely than others, possibly modulated by factors such as odorant molecular properties or perceptual characteristics (Sinding, Puschmann & Hummel, 2014; Dickmänken et al., 2024).

The Sniffin’ Sticks battery, developed in Germany, is a widely adopted psychophysical method for evaluating human olfaction (Kobal et al., 1996; Hummel et al., 1997; Rumeau, Nguyen & Jankowski, 2016). The full version assesses olfactory threshold, discrimination, and identification. Its 16-item identification subtest (SS-16) offers a rapid screening tool (<10 min) and possesses diagnostic utility, particularly in neurology for identifying olfactory dysfunction in prodromal (Siderowf et al., 2012; Haehner et al., 2007) and established neurodegenerative conditions like Parkinson’s disease (PD) and other parkinsonian syndromes (Doty, 2012; Katzenschlager & Lees, 2004; Silveira-Moriyama et al., 2008). Notably, olfactory impairment can manifest decades before characteristic motor symptoms in PD (Doty, 2012; Doty, 2012). The SS-16 functions as a semi-objective measure, bridging the gap between subjective patient reports of smell loss and objective testing. Importantly, research indicates a limited correlation between patients’ subjective olfactory complaints and their performance on objective tests like Sniffin’ Sticks, highlighting the value of psychophysical assessment (Molnár et al., 2023; Nguyen, Nguyen-Thi & Jankowski, 2012).

Because odor familiarity and descriptor semantics vary across cultures, local adaptation and validation are essential prerequisites for their reliable use (Antsov et al., 2014; Silveira-Moriyama et al., 2009; Neumann et al., 2012; Donnabella Bastos et al., 2015; Schriever et al., 2018; Nakanishi et al., 2024). Previous Brazilian studies used adapted translations (Donnabella Bastos et al., 2015; Santin et al., 2010; Trentin et al., 2022). The SS-16 used in this study is based on the standard version, with descriptors translated and adapted into Brazilian Portuguese. Previous work by Silveira-Moriyama et al. (2008) utilized a similar Brazilian Portuguese adaptation of the SS-16 descriptors (e.g., ‘turpentine’ translated as ‘solvente de tinta’, ‘peppermint’ as ‘menta’) and demonstrated its utility in distinguishing Brazilian PD patients from controls.

While prior Brazilian studies have employed adapted translations, none have specifically utilized item-response theory, such as Rasch analysis, to examine the psychometric properties of the SS-16 within a well-characterized, homogeneous adult cohort, nor focused on item-level demographic effects within such a sample. Large normative datasets consistently demonstrate effects of biological sex, age, and education on olfactory performance, underscoring the need to investigate these factors in specific populations (Hummel et al., 2007; Oleszkiewicz et al., 2019; Nakanishi et al., 2024; Fornazieri et al., 2015; Fornazieri et al., 2019).

Accordingly, this study aimed to: (i) evaluate the internal structure validity of the Brazilian Portuguese SS-16 adaptation by examining its unidimensionality using Rasch modeling; (ii) quantify the independent contributions of sex, age, and education to overall and item-specific SS-16 performance (validity evidence based on relations to other variables); and (iii) establish preliminary reference thresholds for classifying olfactory function in a sample of highly educated Brazilian healthcare professionals.

Materials and Methods

Study design and participants

This prospective cross-sectional study received ethical approval from the Institutional Review Boards of Sírio-Libanês Hospital and the University of Brasília (CAAE 31378820.1.2004.5461). All participants provided written informed consent prior to enrollment. The research adhered to the principles outlined in the Declaration of Helsinki and its subsequent amendments regarding research involving human subjects.

Healthcare professionals (e.g., nurses, nurse technicians, physicians, physician assistants, pharmacists, physical therapists, health administrators, psychologists, dentists, and nutritionists) employed at Hospital Sírio-Libanês (Brasília, Federal District, Brazil) were recruited between August and November 2020 through non-probabilistic convenience sampling through internal institutional advertising. Inclusion criteria specified participants to be at least 18 years old and in stable general health. Exclusion criteria encompassed neurological disorders, history of olfactory dysfunction requiring medical attention, autoimmune diseases, endocrine disorders affecting smell (uncontrolled Type 2 diabetes, adrenal or thyroid dysfunction), morbid obesity (body mass index >35), acute rhinosinusitis symptoms, illiteracy, or any current/past SARS-CoV-2 infection confirmed through clinical history, RT-qPCR, or serology. The researchers acknowledged a limitation in their methodology, noting that comprehensive otorhinolaryngological examinations (e.g., nasal endoscopy, Computed Tomography (CT) imaging) were not conducted due to logistical constraints and ethical considerations regarding invasive procedures in asymptomatic participants, meaning subclinical sinonasal conditions potentially affecting olfactory airflow could not be definitively ruled out. Although objective Ear, Nose, and Throat (ENT) examination (e.g., nasal endoscopy) was not performed, this approach aligns with international olfactory testing guidelines, which permit symptom-based exclusion criteria in large-scale or non-interventional olfactory research when ethical or logistical limitations are present (Patel et al., 2022).

Experimental procedures for the assessment of olfaction, SARS-CoV-2 PCR, and serological testing

Olfactory testing

A summary of the study experimental procedures is provided in Fig. 1. The SS-16 kit (Burghart Messtechnik GmbH, Holm, Germany) was administered by trained health professionals (research assistants, nurses) under supervision of a board-certified neurologist (PRPB). This test employed 16 odorants presented via felt-tip pens. Participant responses were elicited using a forced-choice method, offering four Brazilian Portuguese verbal descriptor options per odorant (Table 1) based on the adaptation used by Silveira-Moriyama et al. (2008).

Figure 1 Experimental procedures.

The SS-16 kit was administered by trained health professionals under supervision of a board-certified neurologist (PRPB). All study participants underwent RT-qPCR testing for SARS-CoV-2 from nasal swab samples. Viral RNA was extracted using the QIAamp Viral RNA Mini kit and QIAcube Extractor. Detection employed the BIOMOL OneStep/COVID-19 kit. IgA and IgG antibodies against SARS-CoV-2 were measured by ELISA.

Table 1 Olfactory, RT-qPCR and serological testing.

The SS-16 kit was administered by trained health professionals (research assistants, nurses) under supervision of a board-certified neurologist (PRPB). All study participants underwent RT-qPCR testing for SARS-CoV-2 from nasal swab samples, and IgA and IgG antibodies against SARS-CoV-2 were measured by ELISA.

SS-16 item	Odor descriptors	SS-16 item	Odor descriptors	
1	laranja, morango, amora, abacaxi
(orange, strawberry, blackberry, pineapple)	9	cebola, alho, repolho, cenoura
(onion, garlic, cabbage, carrot)	
2	fumaça, couro, cola, grama (smoke, leather, glue, grass)	10	cigarro, vinho, café, fumaça
(cigarette, wine, coffee, smoke)	
3	mel, chocolate, baunilha, canela
(honey, chocolate, vanilla, cinnamon)	11	melão, laranja, pêssego, maçã
(melon, orange, peach, apple)	
4	cebolinha, “pinho-sol”, menta, cebola
(chives, pine tree, peppermint, onion)	12	cravo, canela, pimenta, mostarda
(clove, cinnamon, pepper, mustard)	
5	coco, nozes, banana, cereja
(coconut, walnuts, banana, cherry)	13	pera, pêssego, ameixa, abacaxi
(pear, peach, plum, pineapple)	
6	pêssego, limão galego, maçã, laranja lima
(peach, lemon, apple, sweet orange)	14	camomila, rosa, framboesa, cereja
(chamomile, rose, raspberry, cherry)	
7	alcaçuz, menta, cereja, bolacha
(licorice, peppermint, cherry, cookie)	15	anis, mel, pinga, “pinho-sol”
(anise, honey, sugar cane liquor, pine tree)	
8	mostarda, bala de menta, borracha, solvente de tinta
(mustard, mint candy, rubber, turpentine)	16	pão, queijo, peixe, presunto
(bread, cheese, fish, ham)	
Note:

Legend: Odor descriptors (target response and distractors) are described in Brazilian Portuguese and correspondent translation to English. Correct responses are highlighted in bold. SS16, Sniffin´ Sticks Identification Test (16 items).

Testing was conducted using the ‘Odor-Lines-on-Paper’ technique as a precaution to minimize potential aerosol generation and infection risk during the COVID-19 pandemic. For each item, an odor line (~5 cm) was drawn on a fresh sheet of paper by the administrator, and the paper was presented approximately 2 cm from the participant’s nostrils for 3 s. Participants then selected one of the four descriptors. A minimum inter-stimulus interval of 20 s was maintained to prevent olfactory fatigue. Participants also completed an electronic questionnaire gathering demographic data (age, sex, education level), clinical comorbidities, smoking history, and self-reported history of olfactory symptoms.

SARS-CoV-2 PCR

All study participants underwent reverse transcriptase quantitative polymerase chain reaction (RT-qPCR) testing for SARS-CoV-2 from nasal swab samples. Nasopharyngeal swabs were collected using rayon-tipped swabs, which were inserted into the nasal passage, twisted 5–10 times, left in place for 15–30 s, and stored in a transport medium. Viral RNA was extracted using the QIAamp Viral RNA Mini kit and QIAcube Extractor (Qiagen, Hilden, Germany), adhering to the provided manufacturer’s protocol. Detection employed the BIOMOL OneStep/COVID-19 kit (Institute of Molecular Biology of Paraná–IBMP, Brazil), which targets the SARS-CoV-2 orf1ab and nucleocapsid genes, employing the CFX 96 TOUCH 1000TM, a Bio-Rad PCR System (Hercules, CA, USA).

Serological testing

IgA and IgG antibodies against SARS-CoV-2 were measured by ELISA (Anti-SARS-CoV-2 ELISA IgA and Anti-SARS-CoV-2 ELISA IgG; Euroimmun Medizinische Labordiagnostika, Lübeck, Germany). Peripheral blood was drawn from each participant, after which serum samples were separated, partitioned into 1 mL aliquots, and preserved at −80 °C. Thawed samples were analyzed according to manufacturer’s guidelines at the University of Brasília Neurovirology Lab.

Data analysis

Categorical variables were represented as frequency (n) and percentage (%). Depending on the normality of the distribution, assessed by the Shapiro-Wilk test, continuous variables were described as mean and standard deviation (sd) or median and interquartile range (IQR). Data analysis was performed using R software (R Core Team, 2023). All statistical tests were two-sided, with a significance threshold set at 0.05.

Rasch model

The Rasch Measurement Model (specifically, the dichotomous logistic model) was employed to analyze the item response data from the SS-16 instrument. Analysis was performed using Conditional Maximum Likelihood Estimation implemented in the eRm package (version 1.0-2) in R (Padgett & Morgan, 2020; Mair & Hatzinger, 2007). This model positions both persons (participants’ olfactory identification ability, denoted by θ) and items (odorant difficulty, denoted by η) on a common interval scale (logits). The model assumes that the probability of a correct response depends only on the difference between person ability and item difficulty.

Adherence of the data to the model’s assumption of unidimensionality was assessed using Principal Component Analysis of Residuals (PCA-R), evaluating the eigenvalue of the first principal component of the residuals against Linacre’s criterion (eigenvalue <2.0 suggests residuals primarily represent random noise rather than a substantive secondary dimension) (Linacre, 1999). Person-item mapping (Wright map) was generated using plotPImap to visualize the alignment of person abilities and item difficulties (Padgett & Morgan, 2020; Linacre, 2022). Item fit to the Rasch model was evaluated using infit and outfit mean square statistics (MNSQ) and their standardized Z-scores (t-statistics) (Linacre, 2002). MNSQ values ideally approximate 1.0; values substantially >1.3 suggest underfit (more variability than predicted, potentially due to guessing or multidimensionality), while values <0.7 suggest overfit (less variability than predicted, potentially indicating item redundancy). Corresponding t-statistics outside ±1.96 are typically considered indicative of significant misfit.

Impact of age, sex, and education on SS-16 scores

Multiple linear and logistic regression analyses were conducted to evaluate the effects of sex, age, and education on SS-16 performance, with standardized β coefficients and partial η2 values reported as measures of effect size.

A linear regression model was applied to assess the influence of these demographic factors on total SS-16 scores, using ordinary least squares estimation on standardized data. The 95% Confidence Intervals (CIs) and p-values were determined using the Wald t-distribution approximation method. Additionally, item-level logistic regression analyses were performed, with each SS-16 item response (coded as 0 = incorrect, 1 = correct) as the dependent variable and sex, age, and education as independent variables. Model validity was confirmed through systematic evaluation of residual patterns to ensure all regression assumptions were met.

Normative tables for SS-16 total score

A continuous normalization approach was implemented to develop preliminary normative classifications for the SS-16 total scores within this specific sample. Raw scores (sum of correctly identified items) were transformed into standard normal quantiles using the rankit method (inverse normal transformation). This transformation preserves the original percentile ranks while mapping the empirical distribution onto a Gaussian scale (Soloman & Sawilowsky, 2009). To better model the potentially non-linear relationship between raw scores and transformed scores, especially in potentially skewed distributions, splines were incorporated within the modeling process linking raw scores to their normalized equivalents, aiming for more accurate representation across the score range. Percentile ranks corresponding to specific score thresholds were derived from this normalized distribution.

Open science disclosure

All measures, conditions and exclusions are reported. Sample size (n = 144) was determined a priori to satisfy around 10 participants per item for stable Rasch parameter estimates. The anonymized dataset is available on Zenodo (https://zenodo.org/records/15384922).

Results

Participants

Out of an initial 196 healthcare professionals recruited, 50 individuals (25.5%) were excluded due to evidence of prior or current SARS-CoV-2 infection (positive IgG or IgA ELISA). Additionally, two participants (1.0%) were excluded due to positive SARS-CoV-2 RT-qPCR results at the time of testing, despite being asymptomatic. No participants were excluded based on other criteria. This resulted in a final cohort of 144 participants, whose ages varied from 18 to 62 years (mean 34.0 ± 8.4 years). Table 2 provides a comprehensive overview of the demographic and clinical characteristics of the participants, including educational levels and clinical features. The majority of the participants were women (n = 101, 71%), reflecting the known female predominance in the healthcare professions, particularly within hospital-based roles typical of this study’s recruitment setting (Pérez-Sánchez, Madueño & Montaner, 2021; Boniol et al., 2019). A history of tobacco exposure (current or former smoker) was reported by 29 participants (20.1%), a figure that aligns with previous studies conducted in Brazil (Doty & Cameron, 2009). Additionally, 52 participants (36.1%) reported a history of chronic rhinitis, presumed to be primarily allergic in nature given the exclusion of acute rhinosinusitis. The mean total SS-16 score for the sample was 13.56 (SD = 1.47).

Table 2 Overview of the demographic and clinical characteristics of the participants.

Feature	Total (n = 144)	
Age, y, mean ± sd	34.0 ± 8.4	
Age classification, n (%)	
Young adult (18–34 y)	82 (56.9%)	
Early middle-aged adult (35–50 y)	54 (37.5%)	
Late middle-aged adult (50–65 y)	8 (5.6%)	
Female Sex, n (%)	101 (70.1%)	
Education, y, mean ± sd	14.8 ± 5.4	
Educational attainment level, n (%)	
Primary	24 (16.7%)	
Secondary	19 (13.2%)	
Tertiary (incomplete or complete Higher Education)	101 (70.4%)	
Comorbidities, n (%)	
Allergic rhinosinusitis	53 (36.8%)	
Essential hypertension	12 (8.3%)	
Diabetes mellitus	3 (2.1%)	
Lung disorder (asthma/COPD)	4 (2.8%)	
Hypothyroidism	5 (3.5%)	
Note:

sd, standard deviation; y, years.

Item-level Rasch analysis

The Principal Component Analysis of Residuals (PCA-R).

yielded eigenvalues for the first contrast below Linacre’s threshold of 2.0, supporting the assumption of unidimensionality. Figure 2 present the Person-item (Wright) map for the SS-16, aligning participant abilities and item difficulties along a singular dimension.

Figure 2 Person-item (Wright) map for the SS-16, aligning participant abilities and item difficulties along a singular dimension.

Variations in the identification rates of different odors were observed across the SS-16 pens, as outlined in Table 3. Most items were concentrated within the range of +1 to −1 logit on Wright’s map, enhancing the accuracy of estimating participants’ theta values within this interval.

Table 3 Item correctly identified in the smell identification test from Sniffin’ sticks (SS-16).

SS-16 item	Individuals who correctly identified the item (%)	SS-16 item	Individuals who correctly identified the item (%)	
laranja (1), orange	144 (100%)	anis (15), anise	121 (84%)	
menta (4), peppermint	141 (97.9%)	couro (2), leather	119 (82.6%)	
canela (3), cinnamon	140 (97.2%)	abacaxi (13), pineapple	116 (80.6%)	
peixe (16), fish	140 (97.2%)	alho (9), garlic	115 (79.9%)	
cravo (12), cloves	133 (92.4%)	café (10), coffee	113 (78.5%)	
banana (5), banana	132 (91.7%)	alcaçuz (7), liquorice	109 (75.7%)	
rosa (14), rose	123 (85.4%)	solvente de tinta (8), turpentine	104 (72.2%)	
limão galego (6), lemon	122 (84.7%)	maçã (11), apple	81 (56.2%)	
Note:

Data present as counts (percentage). The number in the parenthesis after the odor name represents the display order during SS-16 testing. The words in italics represent the original odor descriptors in English.

The odors “menta” (peppermint; η = −1.93 logits, 95% CI [−3.01 to −0.85]), “canela” (cinnamon; η = −1.63 logits, 95% CI [−2.57 to −0.69]), and “peixe” (fish; η = −1.63 logits, 95% CI [−2.57 to −0.69]) were the least difficult to recognize, with correct identifications by 97.9%, 97.2%, and 97.2% of individuals, respectively. Conversely, “maçã” (apple; η = 1.71 logits, 95% CI [1.35–2.05]), “solvente de tinta” (turpentine/paint thinner; η = 0.98 logits, 95% CI [0.61–1.37]), and “alcaçuz” (liquorice; η = 0.80 logits, 95% CI [0.41–1.19]) were the most difficult, with only 56.2%, 72.2%, and 75.7% of correct answers, respectively. These latter items, therefore, may serve as more effective discriminators in identifying individuals with superior olfactory identification skills. The item “laranja” (orange) was excluded from the Rasch analysis due to invariant responding (100% correct identification). The remaining 15 items exhibited acceptable cultural appropriateness, being correctly identified by at least 56% of the sample (Silveira-Moriyama et al., 2009).

Infit (inlier-sensitive fit) t-statistics for the items, excluding “laranja” due to the aforementioned reason, indicated that most items were within the acceptable range of −1.96 to 1.96. As detailed in Table 4, item 11 (“maçã”/apple) displayed the most atypical response pattern, with both infit and outfit t-statistics exceeding 2.0 (Infit MNSQ = 1.63, t = 2.9; Outfit MNSQ = 1.86, t = 2.4), suggesting its response pattern deviated significantly from the model’s expectations. This indicates potential issues with this item’s functioning within the scale for this sample.

Table 4 Difficulty estimates and performance deviations (infit and outfit) on the SS-16 items.

Item	chi-square	df	p-value	Outfit MSQ	Infit MSQ	Outfit t	Infit t	
maçã (11), apple	160.663	134	0.058	1.190	1.164	2.483	2.567	
café (10), coffee	116.164	134	0.865	0.860	0.911	−0.873	−0.807	
solvente de tinta (8), turpentine	124.336	134	0.714	0.921	0.948	−0.617	−0.580	
couro (2), leather	163.121	134	0.044	1.208	1.062	1.091	0.507	
banana (5), banana	111.468	134	0.922	0.826	0.875	−0.481	−0.502	
alcaçuz (7), liquorice	134.254	134	0.478	0.994	1.029	0.008	0.325	
cravo (12), cloves	116.204	134	0.864	0.861	0.906	−0.327	−0.324	
limão galego (6), lemon	102.243	134	0.981	0.757	0.948	−1.195	−0.315	
rosa (14), rose	118.532	134	0.827	0.878	0.956	−0.514	−0.245	
abacaxi (13), pineapple	131.010	134	0.557	0.970	0.974	−0.115	−0.177	
anis (15), anise	127.987	134	0.630	0.948	1.017	−0.193	0.166	
menta (4), peppermint	64.180	134	1.000	0.475	0.854	−0.727	−0.125	
alho (9), garlic	141.752	134	0.307	1.050	1.002	0.349	0.057	
canela (3), cinnamon	168.406	134	0.024	1.247	0.954	0.585	0.040	
peixe (16), fish	116.052	134	0.866	0.860	0.921	−0.052	−0.037	
Note:

MSQ, Mean Square; df, degrees-of-freedom; SS-16, Sniffin’ Sticks odor identification test with 16 items. Observation: the item “orange” could not be included in this analysis due to the lack of variability. All items had an infit and outfit MSQ between the interval of 0.7–1.3.

Influences of gender, age, and educational background on SS-16 scores

Multiple regression analyses were conducted to examine the influence of sex, age, and education on SS-16 performance. A multiple linear regression model predicting the total SS-16 score from sex (0 = female, 1 = male), age (years), and education (years) was statistically significant (F(3, 140) = 2.80, p = 0.042), explaining a small portion of the variance (R2 = 0.056, Adjusted R2 = 0.036). Detailed results are presented in Table 5. The analysis revealed a significant effect of sex, with male participants scoring significantly lower than female participants (β = −0.67, 95% CI [−1.19 to −0.14], t(140) = −2.52, p = 0.013; Standardized β = −0.21). Neither age (β = −0.02, 95% CI [−0.04 to 0.01], t(140) = −1.08, p = 0.284; Standardized β = −0.09) nor education (β = −0.002, 95% CI [−0.05 to 0.04], t(140) = −0.09, p = 0.932; Standardized β = −0.007) showed a statistically significant association with the total SS-16 score in this model. To explore finer-grained effects, separate multiple logistic regression models were fitted for each of the 15 SS-16 items included in the Rasch analysis (excluding ‘orange’). Each model predicted the probability of a correct response (1 = correct, 0 = incorrect) based on sex, age, and education. Detailed results, including odds ratios (OR) and 95% CIs, are presented in Table 5. Significant associations were found for specific items: Increasing age was significantly associated with a lower probability of correctly identifying “café” (coffee) (OR per year = 0.95, 95% CI [0.91–0.99], p = 0.021) and “cravo” (cloves) (OR per year = 0.88, 95% CI [0.82–0.95], p = 0.001). Female participants had significantly higher odds of correctly identifying “rosa” (rose) compared to male participants (OR = 3.16 for female vs. male, 95% CI [1.15–8.67], p = 0.026). Higher educational attainment (years) was significantly associated with an increased probability of correctly identifying “cravo” (cloves) (OR per year = 1.13, 95% CI [1.03–1.24], p = 0.011) and “anis” (anise) (OR per year = 1.13, 95% CI [1.03–1.24], p = 0.011). These item-level findings suggest that while age and education did not significantly predict the total score in this relatively young and highly educated sample, they did influence the ability to identify specific odors.

Table 5 SS-16 total score distribution for the health professionals sample.

SS-16 raw score	Percentile	Smell identification capacity category	
<10	<0.4	Very low	
10	0.4–2.2	
11	2.3–9.0	Low	
12	9.1–25.1	Average	
13	25.2–36.8	
14	36.9–63.0	
15	63.1–84.0	
16	84.1–99.9	High	
Note:

Data based on rankit regression.

Distribution of SS-16 total scores

The distribution of total scores for the SS-16 was based on rankit regression analysis, as shown in Table 6. The ability to identify smells was classified into four categories based on the number of correct answers: “very low” (≤10), “low” (11), “average” (12–15), and “high” (16). These categories are aligned with percentile ranges that are derived from the raw scores of the SS-16, which are the sums of correctly identified items.

Table 6 Multiple linear and logistic regression results.

Each model predicted the probability of a correct response (1 = correct, 0 = incorrect) based on sex, age, and education.

Model	Variable	Predictor	β	Std. β	Std. error	95% CI for β	p-value	Effect size	95% CI for effect size	
Linear regression	Total SS-16 score	Intercept	14.33	–	0.57	[13.21–15.44]	<0.001*	–	–	
		Sex (male)	−0.67	−0.45	0.26	[−1.19 to −0.14]	0.013*	η2 = 0.043	–	
		Age	−0.02	−0.09	0.01	[−0.04 to 0.01]	0.284	η2 = 0.008	–	
		Education	−0.002	−0.007	0.02	[−0.05 to 0.04]	0.932	η2 < 0.001	–	
Logistic regression	Coffee	Age	−0.05	–	0.02	[−0.09 to −0.01]	0.021	OR = 0.95	[0.91–0.99]	
	Cloves	Age	−0.12	–	0.04	[−0.20 to −0.05]	0.001*	OR = 0.89	[0.82–0.95]	
	Cloves	Education	0.12	–	0.05	[0.03–0.22]	0.011*	OR = 1.13	[1.03–1.25]	
	Rose	Sex (male)	−1.15	–	0.52	[−2.17 to −0.14]	0.026*	OR = 0.32	[0.11–0.87]	
	Anise	Education	0.12	–	0.05	[0.03–0.22]	0.011*	OR = 1.13	[1.03–1.25]	
Notes:

Linear regression model: R2 = 0.06, F(3, 140) = 2.80, p = 0.042, adj. R2 = 0.04.

β, regression coefficient; Std. β, standardized regression coefficient; Std. error, standard error; CI, confidence interval; η2, partial eta-squared; OR, odds ratio.

For logistic regression, OR < 1 indicates decreased odds of correct identification with increasing predictor values, while OR > 1 indicates increased odds. For example, an OR of 0.95 for age effect on coffee identification means that for each additional year of age, the odds of correctly identifying coffee decrease by approximately 5%.

* Significant at p < 0.05

Discussion

This research investigated the psychometric properties of a Brazilian Portuguese adaptation of the SS-16 (Hummel et al., 1997; Silveira-Moriyama et al., 2008), in a specific cohort of highly educated, young to middle-aged adult Brazilian healthcare professionals. Employing Rasch analysis, the study provides evidence supporting the unidimensionality of the SS-16 in this sample, suggesting its items collectively measure a single underlying latent trait of odor identification ability. The test demonstrated a range of item difficulties, aligning with the Rasch model’s expectation of items spanning a continuum. These findings offer preliminary support for the SS-16 as a potentially viable instrument for rapid olfactory screening in Brazil, although important limitations regarding generalizability must be considered.

The cultural adaptation of olfactory tests is crucial. The SS-16 version used here employed Brazilian Portuguese descriptors adapted previously. This adaptation differs slightly from the European Portuguese version validated by Ribeiro et al. (2016), which replaced several original German descriptors deemed unfamiliar with terms considered more culturally relevant in Portugal. To accommodate cultural differences, several item descriptors were adapted with more regionally familiar terms: ‘gummy bear’ (a small, chewy candy shaped like a bear) was replaced with ‘goma de fruta’ (fruit gum), ‘sauerkraut’ (fermented cabbage with a tangy flavor) with ‘couve’ (cabbage), ‘fir’ (an evergreen tree with a woody, resinous aroma) with ‘pinheiro’ (pine tree), ‘grapefruit’ (a large citrus fruit with a sour and bitter taste) with ‘laranja’ (orange), ‘turpentine’ (a volatile oil derived from pine resin used as a solvent) with ‘diluente de tinta’ (paint thinner), and ‘peppermint’ (a fragrant hybrid mint commonly used for flavoring) with ‘hortelã’ (mint). The Brazilian Portuguese adaptation used similar logic but features slightly different terms: “goma de mascar” (chewing gum), “repolho” (cabbage), “pinho-sol” (a brand of pine cleaner), “laranja” (orange), “solvente de tinta” (paint thinner), and “menta” (mint/peppermint), respectively (Silveira-Moriyama et al., 2008). The present study’s use of ‘solvente de tinta’ for turpentine and ‘menta’ for peppermint aligns with this Brazilian adaptation approach.

Consistent with some previous findings in Brazil and internationally (Santin et al., 2010), ‘turpentine’ (solvente de tinta) and ‘apple’ (maçã) emerged as relatively difficult odors to identify. The difficulty with ‘turpentine’ might reflect varying familiarity across populations, although paint thinner is a relatively common household and occupational substance (Hedner et al., 2010; Santin et al., 2010). The ‘apple’ item exhibited particularly poor fit statistics (infit/outfit MNSQ > 1.6, t > 2.0) alongside high difficulty (η = 1.71). This misfit suggests inconsistent response patterns not well explained solely by the participants’ overall olfactory ability. This anomalous performance warrants careful consideration. It may stem from cognitive factors beyond pure olfactory detection. The specific apple odorant used might be perceptually similar to distractor options (‘melão’ [melon], ‘laranja’ [orange], ‘pêssego’ [peach]), creating a high cognitive load during the forced-choice task (Hoffmann-Hensel et al., 2017). Furthermore, accurately linking the percept to the label ‘maçã’ requires effective semantic retrieval (Hedner et al., 2010; Olofsson & Gottfried, 2015; Donnabella Bastos et al., 2015). Difficulty distinguishing between similar fruity scents or accessing the specific label could lead to inconsistent responses among individuals with similar underlying olfactory function, thus violating Rasch model expectations (Linacre, 1999). While challenging items can be valuable for assessing cognitive aspects of olfaction, significant misfit raises concerns about the item’s contribution to reliable measurement of the intended unidimensional construct. Future adaptations might consider revising the distractor list for ‘apple’, as done in some pediatric versions (SS-16-Child) (Donnabella Bastos et al., 2015; Bastos, 2018), or further investigating its psychometric behavior in diverse Brazilian samples. The high difficulty of ‘liquorice’ (alcaçuz) is also consistent with findings in other non-Northern European populations, likely reflecting lower cultural familiarity with this specific flavor/scent profile (Konstantinidis et al., 2008; Millar Vernetti et al., 2016; Choi et al., 2024; Zhang et al., 2023).

The regression analyses confirmed the well-established female advantage in olfactory identification, observed here as a significant effect on the total SS-16 score (Sorokowski et al., 2019) Item-level analyses provided more specific insights. The finding that women significantly outperformed men in identifying ‘rose’ may reflect both biological factors (e.g., potential hormonal influences on receptors sensitive to floral compounds like phenylethyl alcohol (Pfabigan et al., 2022)) and sociocultural factors (e.g., potentially greater lifelong exposure and familiarity with floral scents in products typically marketed towards women).

Age showed a significant negative association with the correct identification of ‘coffee’ and ‘cloves’ at the item level, even though no significant effect of age was found for the total SS-16 score. The lack of an overall age effect on the total score is likely attributable to the study’s age range (18–62 years) and not including older adults (>62 years). Significant declines in overall olfactory identification ability are typically observed starting around age 60–65 in large European normative samples (Hummel et al., 2007; Oleszkiewicz et al., 2019). Our findings suggest that subtle, odorant-specific declines may begin earlier in adulthood. The specific vulnerability of coffee and clove identification to aging potentially indicates differential age-related changes in the sensitivity or number of olfactory receptors responsive to the key volatile compounds in these substances (e.g., thiols, guaiacols, eugenol).

Education level significantly predicted better identification of ‘cloves’ and ‘anise’. This finding supports the hypothesis that identifying certain complex or specialized aromas relies more heavily on cognitive-semantic processing (Hoffmann-Hensel et al., 2017; Hedner et al., 2010; Larsson et al., 2004), which is often correlated with educational attainment (Nakanishi et al., 2024). ‘Cloves’ and ‘anise’ possess distinct aromatic profiles often associated with specific culinary or cultural contexts. Recognizing and correctly labeling them may require more developed semantic networks and vocabulary related to spices and flavors, potentially acquired through formal education or related life experiences. However, the impact of education might have been attenuated in this study due to the sample’s homogeneity–most of participants had high levels of formal education. This likely created a ceiling effect, reducing the variance in education and thus the statistical power to detect broader effects on the total score or other items. A sample with greater educational diversity would be needed to fully explore the relationship between education and olfactory identification in the Brazilian context.

The normative classification proposed (e.g., ≤11 correct indicating performance below the 10th percentile, potentially “functionally hyposmic”) provides a preliminary benchmark for this specific demographic. This P10 cutoff aligns reasonably well with thresholds used in large European and Asian studies employing the SS-16 or similar tests (Hummel et al., 2007; Oleszkiewicz et al., 2019). However, defining functional hyposmia based solely on population percentiles requires caution. Thomas Hummel, a co-creator of the test, notes that categorizing a person as functionally hyposmic is somewhat subjective, as it relies on comparing their olfactory function to the normative data of a specific age group (Hummel et al., 2007). Furthermore, it is crucial to acknowledge that individuals possess varying baseline olfactory abilities. Some may have exceptional sensitivity (‘hyperosmia’), and for them, a decline in function might be subjectively perceived even if their score remains within the ‘normosmic’ range according to population norms (Molnár et al., 2023). Clinical interpretation should ideally consider individual baseline function if known, although this is often not feasible.

Clinical utility and implementation

The Brazilian version of the SS-16 demonstrates significant clinical potential, despite requiring more extensive validation. Its short administration time (less than 10 min) facilitates its use as a screening tool in high-demand clinical settings, both in neurology (for early detection of neurodegenerative risk) and otorhinolaryngology (in the assessment of post-viral or sinonasal olfactory dysfunctions). The preliminary cutoff of ≤11/16 (≤P10 in this sample) may serve as an initial indicator of hyposmia in highly educated adults aged 18–62 years, signaling the need for additional investigation or monitoring.

For effective implementation in the Brazilian healthcare system, several strategic steps are recommended: development of standardized training for healthcare professionals; establishment of comprehensive normative data through multicenter studies that address the country’s regional, socioeconomic, educational, and age diversity; integration of standardized olfactory assessment into national clinical guidelines for relevant conditions; incorporation of the SS-16 into existing protocols for suspected neurodegenerative diseases or persistent post-viral symptoms; and addressing potential barriers such as costs, kit availability, need for regional calibration, and clinical time constraints through educational initiatives that emphasize the value of objective olfactory assessment.

Limitations

This study presents several significant limitations affecting the generalizability of its findings. The sample homogeneity represents a primary constraint, as participants consisted exclusively of healthcare professionals from a single urban center (Brasília), characterized by high educational attainment and substantial female predominance (70.1%). This demographic profile limits the applicability of preliminary norms to Brazil’s broader, more diverse population, which exhibits considerable variation in education, socioeconomic status, sex distribution across age groups, and geographic/cultural backgrounds. The observed performance metrics may exceed general population averages due to these sample characteristics, potentially leading to misclassification when applying these norms to individuals with lower educational levels or from different regions.

Additionally, the absence of systematic otorhinolaryngological evaluations (e.g., nasal endoscopy or imaging) limits the ability to definitively rule out subclinical nasal or sinus conditions that could contribute to conductive olfactory impairment. Nonetheless, given that participants were asymptomatic healthcare professionals with likely heightened health literacy, the risk of undetected obstructive pathology is reduced. Importantly, the reliance on symptom-based screening in such populations is consistent with international consensus guidelines, which support the omission of routine ENT examination in non-interventional or large-scale olfactory studies where ethical or logistical constraints apply (Patel et al., 2022).

The study also lacks assessment of test-retest reliability for the Brazilian Portuguese SS-16, omitting an aspect of psychometric validation that should be addressed in future research through retesting after a standard 2–4 week interval. Furthermore, Brazil’s substantial cultural and geographical diversity suggests that odor familiarity and naming conventions may vary significantly between regions (North vs. South, urban vs. rural), variations this single-center study could not capture. The “Odor-on-Lines” method employed for pandemic safety might introduce subtle differences in stimulus delivery compared to direct pen presentation; although likely minor, potential variations in perceived intensity or consistency relative to standard administration cannot be entirely excluded without direct comparison. Future research should prioritize recruitment of larger, more demographically and geographically diverse samples across Brazil, incorporation of objective otorhinolaryngological assessment, and evaluation of test-retest reliability as essential steps toward comprehensive validation of the SS-16 for widespread Brazilian clinical implementation.

Conclusions

This research provides valuable contributions about psychometric data on the Brazilian Portuguese adaptation of the SS-16 in a cohort of highly educated healthcare professionals from Brasília. The findings demonstrate that the SS-16 exhibits structural validity through Rasch analysis supporting unidimensionality, with item-level analyses revealing significant influences of sex, age, and education on specific odors (rose, coffee, cloves, anise), reinforcing the complex interplay between biological and sociocultural factors in olfactory perception.

The Brazilian version shows promising clinical utility, with its brief administration time (under 10 min) facilitating its use as a screening tool in high-demand clinical settings in both neurology and otorhinolaryngology. The preliminary cutoff of ≤11/16 (≤P10 in this sample) may serve as a starting point to indicate hyposmia in highly educated adults aged 18–62 years, though these normative classifications must be interpreted cautiously given the sample’s demographic constraints. However, significant limitations affect the generalizability of these findings, primarily the sample homogeneity (highly educated, predominantly female, single urban center) and the absence of objective otorhinolaryngological assessment and test-retest reliability data.

In conclusion, while this study lends preliminary support for the SS-16’s structural validity and potential clinical utility as an olfactory screening tool in Brazil, substantial further research is imperative. Future investigations must include larger, nationally representative samples encompassing diverse demographics (age, sex, education, socioeconomic status) and geographic regions to establish robust Brazilian norms and fully validate the SS-16’s cross-cultural applicability. Addressing methodological limitations, such as including objective ENT exams and assessing test-retest reliability, will be crucial for building confidence in the SS-16 as a standard olfactory assessment tool in Brazilian clinical and research settings.

Supplemental Information

Supplemental Information 1 Sniffin’ sticks data.

Supplemental Information 2 STROBE checklist.

Supplemental Information 3 Demographic data of the individuals in the study.

We thank Dr. Thomas Hummel for insightful comments and suggestions that lead to improve the quality of this manuscript.

We thank the board of directors from Hospital Sírio-Libanês and Hospital das Forças Armadas for their support. We thank Laboratório Fleury for the disposal of exam collection kits and disposable material.

We thank Jéssica Lorrane Barbosa da Silva, Ingrid Lemes, Luiz Alexandre Farias de Mello, Erika Faria da Silva, Roberta Danielle Mendonça De Melo Fiuza, Corina Alves Carvalho, Mayra Ramos Lacerda for their support with the exams and data collection.

Additional Information and Declarations

Competing Interests

The authors declare that they have no competing interests.

Author Contributions

Pedro Renato de Paula Brandão conceived and designed the experiments, performed the experiments, analyzed the data, prepared figures and/or tables, authored or reviewed drafts of the article, and approved the final draft.

Danilo Assis Pereira analyzed the data, prepared figures and/or tables, authored or reviewed drafts of the article, and approved the final draft.

Diógenes Diego de Carvalho Bispo analyzed the data, prepared figures and/or tables, authored or reviewed drafts of the article, and approved the final draft.

Felipe von Glehn conceived and designed the experiments, authored or reviewed drafts of the article, and approved the final draft.

Laura Silveira-Moriyama conceived and designed the experiments, analyzed the data, authored or reviewed drafts of the article, and approved the final draft.

Simoneide Souza Titze-de-Almeida conceived and designed the experiments, analyzed the data, authored or reviewed drafts of the article, and approved the final draft.

Marcio Nakanishi analyzed the data, authored or reviewed drafts of the article, and approved the final draft.

Felipe Saldanha-Araujo performed the experiments, authored or reviewed drafts of the article, and approved the final draft.

Enrique Roberto Arganaraz performed the experiments, authored or reviewed drafts of the article, and approved the final draft.

Adriana Pinheiro Ribeiro conceived and designed the experiments, performed the experiments, authored or reviewed drafts of the article, and approved the final draft.

Agda Lima dos Santos performed the experiments, authored or reviewed drafts of the article, and approved the final draft.

Ricardo Titze-de-Almeida conceived and designed the experiments, authored or reviewed drafts of the article, and approved the final draft.

Human Ethics

The following information was supplied relating to ethical approvals (i.e., approving body and any reference numbers):

This research was approved by Sírio-Libanês Hospital and University of Brasilia (CAAE 31378820.1.2004.5461)

Data Availability

The following information was supplied regarding data availability:

The raw data is available in the Supplemental Files.

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
