# Peer review of "Sniffin’ sticks smell identification test: exploring measurement properties in an adult Brazilian healthcare professionals sample"

_PeerJ, doi:10.7717/peerj.19733_

## Round 0.1 · original submission · Major Revisions

I have received three thoughtful reviews from highly qualified reviewers, and I have also thoroughly read your piece. While the reviewers are generally positive about the work, all see areas where your reporting could be improved and have questions that should be addressed before the manuscript meets the threshold for publication.

I believe all of the reviewers' feedback is clear and should be relatively easily addressed. However, I would like to add four points:

1. Your introduction is far too terse and does not prepare the reader for the pattern of results that would be expected in such a validation study. The difference you probed for between sexes, age groups, and education levels should have been foreshadowed in the introduction and explained (to the best of our current ability). Why are these differences predicted and how are they generally explained in the literature?

2. Reviewer 1 requests that you only use the term "sex" throughout your manuscript and abandon use of the term "gender." However, this choice should be contingent upon what variable was actually measured in your study. Were participants invited to report "sex" or "gender?" Stick with the variable name that was actually collected throughout the manuscript.

3. Reviewer 2 has asked you to move the "Participants" section from Results to Methods. I am not convinced it needs to be moved. Rather, it needs to be relabeled as "Exclusions" (or something to that effect). However, you should note the number of participants you initially recruited and their demographic makeup in the Method section.

4. I request that you add a statement to the paper confirming whether you have reported all measures, conditions, data exclusions, and how you determined your sample size. You should, of course, add any additional text to ensure the statement is accurate. This is the standard reviewer disclosure request endorsed by the Center for Open Science [see http://osf.io/project/hadz3]. I include it in every review.

I look forward to reading a revised version of the manuscript. Thanks for submitting it to PeerJ.

Reviewer 1 ·

Basic reporting

First, I would like to thank you for reading this manuscript. This investigation focuses on the Brazilian adaptation of the Sniffin’ Stick test. Adapting this test for different populations is crucial, as a lack of adaptation can lead to biased results. Thus, the topic of this investigation is indeed significant. However, there are several issues that need to be addressed before this manuscript can be considered for publication. Below, I have outlined my specific recommendations for improvement.
Abstract
First, olfactory assessments are essential not only for patients with neurological disorders but also in otorhinolaryngological practice. This was particularly evident during COVID-19, when there was a notable increase in upper-airway infections leading to olfactory disorders.
70% female - The unequal distribution of sex, which indicates a significant dominance of females, should be highlighted as a limitation that may restrict the generalisation of the results.
COVID-19 is an incorrect term; COVID-19 refers to the disease, while SARS-CoV-2 refers to the virus that causes the infection. Please correct this.
In medical writing, the term ‘sex’ should consistently be used instead of ‘gender’.
‘Age was negatively associated with correct endorsements..’ – Please clarify for readers the practical implications of this result.
‘.. with accurate answers to rose’ – It would be beneficial to rephrase this sentence to say, ’with more accurate answers to rose.’
In the conclusion of the abstract, it would be beneficial to present the current study findings in more detail, particularly by specifying the differences related to sex.
Introduction
Line 58. Anosmia is merely one symptom of olfactory disorders. Other symptoms, such as hyposmia, parosmia, and phantosmia, should also be noted.
Line 60. Once again, otorhinolaryngological issues should also be mentioned.
Lines 64-65. In the context of neurodegenerative disorders, it is important to note that olfactory symptoms may appear decades before the other characteristic symptoms of these conditions.
When discussing the Sniffin’ Sticks test, it is important to emphasize that this method is semi-objective. Previous studies have found no correlation between patients' subjective reports and the results of the Sniffin’ Sticks test. Regarding this, please refer to and cite the following article:
doi: 10.3390/jcm12031041.
Lines 68-69. Given these adaptations, were any required changes made?
Material and methods
Line 96-97. Were these neurological conditions excluded based on neurological examinations or was it determined solely from the case history?
Line 98. Once again, the term SARS-CoV-2 infection is more accurate. Furthermore, using ’case history’ instead of ’anamnesis’ is a better choice.
It is important to clarify whether participants underwent an otorhinolaryngological examination, including nasal endoscopy, before the Sniffin’ Sticks test. This examination is crucial for ruling out potential causes of conductive olfactory loss, such as nasal polyps. Additionally, it is necessary to confirm that systemic disorders, such as diabetes mellitus and other endocrine disorders that could lead to olfactory problems, were excluded.
The experimental procedure section should be divided into subsections, such as ’olfactory testing’, ’SARS-CoV-2 PCR’, and ’serology’. It is also important to indicate who performed the Sniffin' Sticks test, such as a doctor or assistant.
Results
Line 199. Only SARS-CoV-2 infection was the reason for excluding potential participants; were any other exclusion criteria not met?
Line 203. It would be helpful to include the mean and standard deviation values for the participants’ ages.
Lines 205-207. Regarding interpreting this result, it is important to note that female predominance may differ across various healthcare professions and medical specialties.
Lines 210-211. To properly assess chronic rhinosinusitis, it is important to clarify how participants were examined before the Sniffin’ Sticks test, such as through nasal endoscopy and/or CT scans of the nasal cavities and paranasal sinuses. This must be clarified. For instance, the presence of nasal polyps can significantly affect the results of olfactory tests.
Lines 231-233. Given these results, cultural differences should also be considered. For example, some populations might be unfamiliar with the scent of turpentine.
Line 246. The term ’sex’ should be used consistently throughout the text. Additionally, it is advisable to present the results of this subsection in a separate table. This will make it easier for readers to understand the results of the linear and multiple logistic regression models.
Line 253. Instead of using ’Std.’, it should be written as ’Std. error.’
Discussion
Line 286. Providing an English explanation for ’Sauerkraut’ would be beneficial. This also applies to other terms collected in lines 286-290 that lack English explanations.
Lines 296-303. It would be intriguing to offer possible explanations for these results.
Lines 320-322. When interpreting olfactory testing, it is important to consider that some participants have an excellent sense of smell, which leads to lower thresholds and increased smell sensitivity. As a result, those with strong olfactory abilities may report changes in their smell perception compared to their usual baseline, even if objective testing does not indicate any significant differences based on normative data. Considering this information, refer to:
doi: 10.3390/jcm12031041.
Lines 326-329. It is also important to note that future study samples should aim for a more balanced distribution of sexes.
Line 338. It is not recommended to use the contraction ’it’s’ in scientific writing; instead, use ’it is.’
It is crucial to acknowledge other limitations of the study, including the unequal distribution of sexes and the lack of otorhinolaryngological assessments, if they were not conducted. This should be clarified and noted as a necessary limitation.
Tables
Table 1. Please correct ‘tes’ to ‘test’ in the table caption. Additionally, clear English explanations should be provided for each item.
Table 4. It is advisable to highlight statistically significant p-values, for instance, by using an asterisk for emphasis to enhance clarity. Additionally, the significance level should be indicated in the table caption.

I am looking forward to receiving the revised version of the manuscript, which includes a point-by-point response to each review comment with all required changes accurately made. This is necessary for deciding whether this manuscript can be considered for publication.

Experimental design

See my specific recommendations above.

Validity of the findings

See my specific recommendations above.

Reviewer 2 ·

Basic reporting

L. 96: The exclusion criteria are rather limited. In fact, it is known that many other factors are associated with olfactory dysfunction. Without going into specific pathologies, autoimmune diseases and increased body weight can be mentioned. Why did the authors not consider them?

L. 246: I think it would be much easier for the reader if these results were represented in a table. Also, it is not clear to me which of the independent variables is more important and which contributes less to the model.

I think the "Participants" paragraph should be moved from the Results section to the M&M section.

Experimental design

L. 110-111: Are the authors sure that the stimulation was actually effective?

L. 271-272: from what was written previously, I had intuited that the smell of orange had been eliminated from the analyses, as it was recognized by 100% of the participants.

L. 305-306: Previously it was written that the maximum age of participants is 62 years. How do authors reconcile these two things?

Validity of the findings

I think the conclusions are a bit weak and vague. The authors should write more directly what can be deduced from their results, especially with a view to future studies.

L. 292: the aim is precisely to evaluate the ability to identify even complicated stimuli, to evaluate, among other things, cognitive abilities and semantic memory.

L. 334-335: the authors have a sample that broadly covers at least two age groups, so they could evaluate at least these ages.

Additional comments

I think that the manuscript of de Paula Brandao entitled “Sniffin' sticks smell identification test: exploring measurement properties in an adult Brazilian healthcare professionals sample” aimed at investigating the validity of the 16-item Sniffin’ Sticks odor identification test (SS-16) in a homogeneous sample of highly educated young and middle-aged Brazilian adults, has only partially achieved its intended purpose.

In fact, in the discussion section, the authors partly repeat the results, but they do not try to find explanations for these results other than cultural differences. I believe that they should try to provide more consistent interpretations, even speculative ones if we want. It is as if the final "therefore" was missing.

Reviewer 3 ·

Basic reporting

This study provides a timely and valuable contribution to olfactory research by validating a culturally adapted version of the 16-item Sniffin’ Sticks odor identification test (SS-16) in a highly educated, young-to-middle-aged Brazilian cohort. The study addresses the critical need for culturally adapted olfactory assessment tools, particularly given the increased focus on olfactory evaluation following the COVID-19 pandemic.

Experimental design

A prospective observational cross-sectional study was conducted. Using Rasch modeling to confirm the unidimensionality of the SS-16 and evaluate item-level performance adds robust statistical rigor to the study. Rigorous exclusion of participants with COVID-19-related olfactory impairments, rhinosinusitis, or neurological disorders ensures the reliability of the findings. Adjusting odor descriptors to align with Brazilian cultural and linguistic norms enhances the test’s applicability to the target population.

The study does not include test-retest reliability assessments, which are crucial for establishing the stability of the SS-16 over time. Can the authors please address this issue?

Recruiting a homogeneous group (healthcare professionals) introduces selection bias, limiting the diversity of perspectives and reducing the study's generalizability. The study does not include participants from rural or less urbanized regions of Brazil, potentially overlooking variations in odor familiarity and recognition.

Validity of the findings

The results section is rigorous, with a clear description of the sample and a detailed analysis using Rasch modeling to confirm the unidimensionality of the SS-16. Identifying items like peppermint, cinnamon, and fish as easiest and apple, turpentine, and liquorice as most challenging is consistent with previous findings in other populations, enhancing the external validity of the item-level performance.The categorization of scores into "very low," "low," "average," and "high" offers a useful framework for clinical application, though further explanation of its practical implications is needed. The discussion contextualizes the findings within broader literature, confirming well-established patterns such as age-related declines and gender differences in olfactory performance. However, the study’s generalizability is limited by the homogeneity of the sample, predominantly composed of highly educated female healthcare professionals, and the lack of representation from diverse educational, geographical, and cultural backgrounds. While the authors acknowledge this limitation and propose directions for future research, a more thorough exploration of these constraints would enhance the discussion.

Additionally, the influence of education on olfactory performance is underexplored, possibly due to ceiling effects, and certain item-specific issues, such as the atypical response pattern for "apple," could benefit from further analysis. Overall, this research underscores the importance of culturally sensitive adaptations in olfactory assessments and highlights the need for more inclusive studies to establish norms that better reflect Brazil's diverse population. With clearer reporting and expanded implications for clinical practice, this study has significant potential to advance the field of olfactory evaluation.

Please, if nothing else, address the following points

Adapting the test to Brazilian Portuguese and acknowledging cultural differences in odor naming and recognition increase its face validity for the specific population studied.

The high educational attainment in the sample may obscure the impact of education on olfactory identification, reducing the construct validity when applied to populations with broader educational variability. Potential regional differences in odor recognition within Brazil are not accounted for, which may affect the test’s cultural validity across the country. It would be important for authors to address clearly in the limitation section

Please explain the following: "The response anomalies for items like "apple" are noted but not investigated further, which could have provided more insights into cultural or semantic factors affecting performance."

While some p-values are reported, the absence of confidence intervals or effect sizes limits the statistical transparency. Including these metrics would give readers a more comprehensive understanding of the data.

Additional comments

While the score classifications are useful, the discussion does not fully address how these thresholds could translate into clinical or public health applications.. Please write a section about clinical utility and relevance of the tool and plans for implementation

---

## Round 0.2 · accepted · Accept

Two of the previous reviewers examined your revision, as did I. We are satisfied that you have addressed the prior concerns and commend you for the improvements you have made to the manuscript. I am happy to accept this work for publication in PeerJ. This manuscript is now ready for publication.


Reviewer 1 ·

Basic reporting

Thank you for sending me the revised version of the manuscript. The authors have made significant efforts to enhance its quality, addressing the most important corrections thoroughly. Therefore, from my point of view, it can now be considered for publication.

Experimental design

After implementing the suggested corrections, the experimental design is now accurate.

Validity of the findings

After making the recommended corrections, the information is now accurate.

Reviewer 2 ·

Basic reporting

The authors have significantly improved the introduction, increasing the basic information needed by the reader. The results section, based on the suggestions received, has also been improved.

Experimental design

The experimental design and research were conducted with rigor

Validity of the findings

Although as one of the reviewers noted, the sample had an excessive number of females, I think the results are valid for the Brazilian population. Of course, differences related to sex will have to be considered, but this is always true. Furthermore, in almost all studies the number of females exceeds that of males.

Additional comments

No other comments